# Population Dynamics of *Aedes aegypti* and *Aedes albopictus* in Two Rural Villages in Southern Mexico: Baseline Data for an Evaluation of the Sterile Insect Technique

**DOI:** 10.3390/insects12010058

**Published:** 2021-01-11

**Authors:** Carlos F. Marina, J. Guillermo Bond, Kenia Hernández-Arriaga, Javier Valle, Armando Ulloa, Ildefonso Fernández-Salas, Danilo O. Carvalho, Kostas Bourtzis, Ariane Dor, Trevor Williams, Pablo Liedo

**Affiliations:** 1Centro Regional de Investigación en Salud Pública-INSP, Tapachula, Chiapas 30700, Mexico; gbond@insp.mx (J.G.B.); aynek2001@hotmail.com (K.H.-A.); armando_ulloa@hotmail.com (A.U.); ildefonso.fernandezsl@uanl.edu.mx (I.F.-S.); 2El Colegio de la Frontera Sur (ECOSUR), Tapachula, Chiapas 30700, Mexico; jvalle@ecosur.mx (J.V.); ador@ecosur.mx (A.D.); pliedo@ecosur.mx (P.L.); 3Facultad de Ciencias Químicas, Universidad Autónoma de Chiapas (UNACH), Tapachula, Chiapas 30700, Mexico; 4Facultad de Ciencias Biológicas, Universidad Autónoma de Nuevo León (UANL), San Nicolás de los Garza, Nuevo León 66450, Mexico; 5Insect Pest Control Laboratory, Joint FAO/IAEA Programme of Nuclear Techniques in Food and Agriculture, IAEA Laboratories, 2444 Seibersdorf, Austria; D.Carvalho@iaea.org (D.O.C.); K.Bourtzis@iaea.org (K.B.); 6Instituto de Ecología AC (INECOL), Xalapa, Veracruz 91073, Mexico; trevor.williams@inecol.mx

**Keywords:** vector control, dengue, oviposition traps, baseline study

## Abstract

**Simple Summary:**

*Aedes aegypti* and *Ae. albopictus* are the most important mosquito vectors of dengue, chikungunya and Zika virus. There is growing interest in the control of these vectors using the sterile insect technique in which large numbers of sterilized males are released and compete with wild fertile males for mates. Females that mate with sterile males do not produce viable offspring. A study was performed on the population fluctuations of *Ae. aegypti* and *Ae. albopictus* using egg traps in two rural villages with a history of dengue in Chiapas, southern Mexico. Higher numbers of *Aedes* eggs were recorded in Hidalgo village compared with the village of Río Florido. In contrast, higher number of eggs were collected in areas surrounding Río Florido, compared with those around Hidalgo. *Aedes aegypti* was the dominant species during the dry season and at the start of the rainy season in both villages. *Aedes albopictus* populations were lower for most of the dry season, but increased during the rainy season and became dominant at the end of the rainy season in both villages. *Aedes albopictus* was also the dominant species in the zones of natural vegetation surrounding both villages. We conclude that the efficacy of a program of vector control involving the sterile insect technique could be evaluated in future studies on the isolated mosquito populations in these rural villages, in combination with habitat elimination and appropriate treatment of water sources.

**Abstract:**

Indoor and outdoor ovitraps were placed in 15 randomly selected houses in two rural villages in Chiapas, southern Mexico. In addition, ovitraps were placed in five transects surrounding each village, with three traps per transect, one at the edge, one at 50 m, and another at 100 m from the edge of the village. All traps were inspected weekly. A transect with eight traps along a road between the two villages was also included. Population fluctuations of *Aedes aegypti* and *Ae. albopictus* were examined during 2016–2018 by counting egg numbers. A higher number of *Aedes* spp. eggs was recorded at Hidalgo village with 257,712 eggs (60.9%), of which 58.1% were present in outdoor ovitraps and 41.9% in indoor ovitraps, compared with 165,623 eggs (39.1%) collected in the village of Río Florido, 49.0% in outdoor and 51.0% in indoor ovitraps. A total of 84,047 eggs was collected from ovitraps placed along transects around Río Florido, compared to 67,542 eggs recorded from transects around Hidalgo. Fluctuations in egg counts were associated with annual variation in precipitation, with 2.3 to 3.2-fold more eggs collected from ovitraps placed in houses and 4.8 to 5.1-fold more eggs in ovitraps from the surrounding transects during the rainy season than in the dry season, respectively. *Aedes aegypti* was the dominant species during the dry season and at the start of the rainy season in both villages. *Aedes albopictus* populations were lower for most of the dry season, but increased during the rainy season and predominated at the end of the rainy season in both villages. *Aedes albopictus* was also the dominant species in the zones surrounding both villages. The numbers of eggs collected from intradomiciliary ovitraps were strongly correlated with the numbers of eggs in peridomiciliary ovitraps in both Río Florido (R^2^_adj_ = 0.92) and Hidalgo (R^2^_adj_ = 0.94), suggesting that peridomiciliary sampling could provide an accurate estimate of intradomiciliary oviposition by *Aedes* spp. in future studies in these villages. We conclude that the feasibility of sterile insect technique (SIT)-based program of vector control could be evaluated in the isolated *Ae. aegypti* populations in the rural villages of our baseline study.

## 1. Introduction

*Aedes aegypti* and *Ae. albopictus* are the most important insect vectors of human diseases in tropical urban habitats across the world [1]. In the Americas, *Ae. aegypti* is considered the main vector of dengue [2,3] and is also involved in the rapid spread of the chikungunya and Zika viruses [4,5]. In contrast, the Asian tiger mosquito, *Ae. Albopictus*, can transmit 26 species of arboviruses, including dengue virus [6], although it is considered to be a vector of secondary importance in semi-urban habitats in the Americas [3,6,7].

*Aedes aegypti* entered the American continent from Africa during the slave trade period [8]. Between the late 19th and early 20th century, this species was implicated as the vector of yellow fever and was the target of an eradication program in Mexico [9,10]. This was successful, and Mexico was certified by the Pan American Sanitary Bureau as free of *Ae. aegypti* in 1963 (Novo, 1995). However, in 1965, it re-infested Mexico through the northern border, and in the late 1970s, it was found at the southern border [11], when it was associated with the resurgence of dengue virus in this region [12,13].

*Aedes albopictus* was first reported in the Americas in 1985 in southeastern Texas [14], where it was probably introduced through international trade in vehicle tires from Asia [15]. This species was also reported in Brazil around the same period [16]. A decade later, *Ae. albopictus* was recorded as being widely distributed in the southeastern United States [17] and on the US-Mexico border [18]. Following subsequent reports of its presence at the southern border of Mexico with Guatemala [19], this vector has dispersed and is currently established in 14 states of Mexico, where it usually coexists in sympatry with populations of *Ae. aegypti* [20,21,22,23,24,25,26].

In Mexico and elsewhere in Latin America, control of *Aedes* spp. mainly involves habitat elimination and regular treatment of domestic and peri-domestic water containers with temephos, an organophosphate larvicide [2,27]. Additionally, in the case of dengue outbreaks, residual spraying of houses and nebulization of streets and surrounding areas with pyrethroid insecticides are undertaken in an attempt to further reduce vector populations, often with limited success [28]. However, due to operational failures in removing oviposition sites [2], low penetration of insecticides into houses during nebulization [29], and mosquito resistance to insecticides [30,31,32,33], the control of this vector has not been successful [2]. Moreover, the lack of vaccines for the principal mosquito-transmitted viruses [34] means that innovative and effective vector suppression strategies are urgently required.

The sterile insect technique (SIT) is an alternative method for vector control. SIT is a species-specific, non-polluting, and environmentally friendly method of insect population control. This technique is based on the mass-rearing and systematic release of large numbers of sterile males that compete with wild males for mating with wild females [35,36,37,38]. The implementation of SIT-based vector control programs requires an area-wide integrated approach [36,37]. Effective SIT programs require knowledge of the mating behavior, ecology, and population dynamics of the target population, prior to the release of sterile males [39,40].

In order to validate the efficacy of a SIT program for the suppression of *Ae. aegypti* in Mexico, a pilot project was established. Two rural village communities with similar environmental and social conditions were selected. The first step involved acquiring baseline information on mosquito populations in each of these villages. In our study area, *Ae. aegypti* and *Ae. albopictus* share habitats in rural and peri-urban areas [20,41]. Therefore, this study aimed to characterize the population fluctuations of both species in two pilot rural village communities as baseline information for the implementation of an area-wide integrated vector suppression program using the SIT. We were interested in both vector species, but with the caveat that *Ae. aegypti* has a wider distribution in urban areas and is the main vector of dengue and other vector-borne diseases in Mexico [2,4,5].

## 2. Materials and Methods

### 2.1. Study Sites

This study was conducted in two rural village communities, approximately 14 km from the city of Tapachula, Chiapas State, in southern Mexico. The villages were (i) Río Florido (14°51′41″ N, 19°21′15″ W), with 172 inhabited houses, 789 inhabitants, and an area of 24 hectares, at a distance of 17.6 km from the Pacific Ocean and, (ii) Hidalgo (14°53′04″ N, 92°21′28″ W) with 184 inhabited houses, 697 inhabitants, and an area of 26 hectares at a distance of 18.4 km from the Pacific Ocean (Figure 1). No mosquito control activities were carried out in either village during the study period. The climate in this area is warm subhumid with a dry and a rainy season [42]. The rainy seasons were taken to have begun when precipitation exceeded 100 mm in a 4-week period, which occurred in March–April and lasted until October–November, followed by a dry season from November–December to March–April (Appendix A). The mean daily temperature ranged from 23.6 to 34.0 °C, with an average of 28.2 °C and an annual mean rainfall of 2867 mm. The villages were 4 km apart and were surrounded by cocoa and palm tree plantations, mango orchards, and other fruit trees and field crops, principally maize (Figure 1).

### 2.2. Methods for Mosquito Collections

*Aedes* spp. eggs were collected from houses and transects in both villages using ovitraps (oviposition traps) comprising black plastic containers (10 cm diameter, 20 cm height) that were three-quarters filled with 1 L of dechlorinated tap water, as used in a previous study [41]. A strip of filter paper (5 cm width × 35 cm length) was placed around the inside of each container at water level, as an oviposition substrate. To reduce disturbance, all containers were labeled with information on the National Institute of Public Health (CRISP-INSP) and an instruction to avoid tampering with the trap.

### 2.3. Experimental Design

#### 2.3.1. Sampling in Houses of the Pilot Communities

Sampling of *Aedes* spp. eggs in ovitraps was performed between 6 January 2016 and 28 November 2018, in 15 houses distributed within each village. For this, each village was divided into 15 roughly square blocks, each of 1.6–1.7 hectares in area. This was the maximum number of houses that could be sampled by our small group of researchers and technicians during the sampling period (09.00 and 12:00 h), given that two villages were sampled on each occasion. One house in each block was randomly selected and permission was obtained from the homeowner to place and monitor ovitraps. Two ovitraps were placed in each house, one inside and one outside the house, in protected sites that were likely to be attractive to *Aedes* mosquitoes and away from other possible oviposition sites (used vehicle tires, water tanks, drums, etc.).

#### 2.3.2. Sampling in Surrounding Areas

Five transects were established on the outside of each village, each comprising three sampling points to determine the presence of *Ae. aegypti* in the area of natural vegetation surrounding each village. One ovitrap was located at the outer edge of the village (~15 m from the nearest dwelling), the second trap was placed 50 m away from the first trap, and the farthest trap at 100 m from the first trap. The ovitraps were installed at ground level or hung on a tree branch at a height of no more than 1 m above the ground.

Another transect was set up along the 4 km road between the two villages, comprising eight sampling points to estimate the potential of the road as a route of *Ae. aegypti* dispersal from one village to the next. This was an issue of interest as a few houses were scattered along the road, although all were set back from the road at a distance of ~100 m. A sampling point was established at 250 m from the edge of each village and the remaining six sampling points were separated by an average distance of 500 m along the road. The ovitraps were installed at ground level under tree canopies. This transect was operated from 13 December 2017 to 18 November 2018. Ovitraps in houses and transects were inspected at weekly intervals. The oviposition substrates with *Aedes* spp. eggs were removed and taken to the laboratory. These were replaced with a new strip of filter paper and the water level was adjusted by the addition of dechlorinated tap water to account for evaporation during the previous week.

The air temperature and relative humidity at the moment of sampling (between 09:00 and 12:00 h) were measured using a digital thermometer-hygrometer (HTC-1, Ace Instruments, India). Water temperature in the containers at the moment of sampling was measured using a laboratory thermometer. Precipitation records were obtained from a weather station located at a distance of 6.0–6.8 km from the villages.

### 2.4. Laboratory Processing of Samples

In the laboratory, eggs present on oviposition substrates were examined using a stereomicroscope, counted, and recorded as hatched or unhatched. At the beginning of the study and until 22 March 2017, samples of up to 50 unhatched eggs were randomly selected and placed in water at 38 °C that gradually cooled to room temperature, following established procedures employed in the insectary of the Centro Regional de Investigación en Salud Pública—INSP, Chiapas, Mexico. The numbers of hatched eggs, number of larvae, and unhatched eggs were then counted 48 h later. Subsequently, starting 29 March 2017, we implemented a method to promote embryo development before estimating hatching rate, in order to obtain more reliable data on the prevalence of unhatched eggs. This method consisted of selecting a random sample of up to 150 unhatched eggs that were placed in a humid chamber for 48 h, so that the embryos within the eggs completed their development. The humid chamber consisted of a 1 cm thick layer of wet cotton placed on a sealed plastic tray (39 × 25 × 5 cm). Following 48 h in the humid chamber, a sample of up to 50 eggs was placed in water at 38 °C, and 48 h later, the numbers of larvae and unhatched eggs were counted. The larvae from each sample were reared at 26 ± 1 °C using 4% liquid suspension (*w*/*v*) of rodent diet (LabDiet 5001, PMI Nutrition International LCC, St. Louis, MO) [43]. Adult mosquitoes were examined and identified to species and sex using an identification key [44].

### 2.5. Statistical Analyses

Mixed effects models with a general correlation structure and a negative binomial response specified were fitted to each of the variables measured. The dependent variables were the *Aedes* spp. eggs collected and the adults of *Ae. aegypti* and *Ae. albopictus*, which were recorded weekly and were summed over 4-week periods prior to the analyses. The hypotheses tested were the differences between years, seasons (rainy, dry), location (indoor, outdoor), villages (Hidalgo, Río Florido), and species (*Ae. aegypti*, *Ae. albopictus*). The analyses were performed in R software, v. 3.4.0 [45]. The effects of fixed factors were performed with the likelihood radio test. The fixed factors were year, village, season, and site. The random factor was the houses. The significance of factor effects in mixed models are presented in terms of χ^2^ values (Appendix A). Correlations were performed between seasonal egg counts from intradomiciliary and peridomiciliary ovitraps from each village. This procedure was also performed based on estimated adult abundance in each season for each species.

## 3. Results

The mean air temperature in Río Florido and Hidalgo houses at the moment of sampling in the mid-late morning period, 09.00–12.00 h, was 32.7 ± 0.04 °C (range 22.7–41.6 °C) and 30.2 ± 0.04 °C (range 22.0–42.2 °C), respectively, whereas mean relative humidity was 60.8 ± 0.2% (range 24.0–99.0%) and 73.2 ± 0.2% (range 30.0–99.0%), respectively. Mean water temperature of ovitraps at the moment of sampling in Río Florido was 27.9 ± 0.03 °C (range 22.0–34.0 °C) and was 27.4 ± 0.03 °C (range 22.0–35.0 °C) in Hidalgo (Appendix A).

Mean air temperature in the area surrounding each village was similar or slightly lower than temperatures measured in each village, whereas mean relative humidity was ~15% higher in the area surrounding Río Florido compared to the inhabited zone and was similar in both inhabited and surrounding zones of Hidalgo. This difference likely reflects a higher abundance of natural vegetation (not measured) surrounding Río Florido compared to that surrounding Hidalgo. The mean water temperature of ovitraps in the zones surrounding each village were similar between Río Florido (25.6 ± 0.03 °C) and Hidalgo (26.5 ± 0.04 °C) (range of temperatures: 21.0–33.0 °C). In the transect along the road between the villages, the mean air temperature was 35.2 ± 0.1 °C (range 28.0–43.2 °C) and the mean relative humidity was 57.6 ± 0.9% (range 22.0–97.0%), mean water temperature of ovitraps at the moment of sampling was 29.5 ± 0.1 °C (range 25.0–40.0 °C) (Appendix A). The total annual precipitation was 2925 mm in 2016, 2805 mm in 2017, and 2841 mm in 2018. The mean precipitation was 28.5 mm/month during the dry season and 2828.5 mm/month in the rainy season (Appendix A, Appendix A).

Small numbers of non-*Aedes* species were identified during laboratory rearing of eggs from oviposition traps. These included *Ochlerotatus atropalpus* (total 43 specimens), *Haemagogus equinus* (1959 specimens), *Culex coronator* (148 specimens), C*x. quinquefasciatus* (101 specimens), other *Culex* spp. (234 specimens), and *Limatus durhamii* (903 specimens). The predatory mosquito *Toxorhynchites theobaldi* was also observed in ovitraps on occasions (343 specimens). In all cases, these species were more prevalent in ovitraps placed in the surrounding zone or along the road transect than in ovitraps from houses, but in all cases these species were excluded from the analyses.

### 3.1. Sampling in and Around Houses

In Río Florido village, a total of 165,623 *Aedes* spp. eggs was collected between 2016 and 2018 (Table 1). The highest quantity was collected in 2018 with 80,215 eggs, while numbers were significantly lower during 2016 with 42,918 eggs and 2017 with 42,490 eggs (χ^2^ = 148.9, df = 2, *p* < 0.001). Over 2.5-fold more eggs were collected during the rainy seasons (119,220 eggs) compared to those collected in the dry seasons (46,403 eggs) (χ^2^ = 69.3, df = 1, *p* < 0.001). In total, 55,033 eggs were collected during the 2018 rainy season, which was 1.8-fold more than the numbers recorded in the rainy season of 2016 (30,522 eggs) and 1.6-fold more than recorded in the rainy season of 2017 (33,665 eggs). Likewise, the number of eggs recorded in the 2017–2018 dry season (25,182) was 2-fold more than recorded in 2016 (12,396 eggs) and 2.9-fold more than recorded in the 2016–2017 dry season (8825 eggs). Total annual egg counts were similar in ovitraps placed inside houses or outdoors over the entire period of the study (Table 1).

In Hidalgo village, a total of 257,712 eggs was collected between 2016 and 2018, which was 1.6-fold more eggs than collected in Río Florido in the same period (Table 2). A significantly higher number of *Aedes* spp. eggs (135,707) was recorded in 2018, compared to 51,968 eggs collected in 2016 and 70,037 eggs collected in 2017 (χ^2^ = 148.9, df = 2, *p* < 0.001). Overall, 3.2-fold more eggs (195,961 eggs) were collected in the rainy seasons in Hidalgo compared to a total of 61,751 eggs collected during the dry seasons (χ^2^ = 69.3, df = 1, *p* < 0.001). A higher number of eggs (195,961) was recorded in the rainy season of 2018 compared to the rainy seasons of 2016 (40,162 eggs) or 2017 (56,952 eggs). A greater number of eggs (36,860) were collected in the dry season of 2017–2018 compared to the dry seasons of 2016 (11,806 eggs) or 2016–2017 (8825 eggs). Ovitraps placed outside of houses in Hidalgo had significantly more eggs (total 149,825 eggs) than ovitraps placed indoors (total 107,887 eggs), an effect that was mainly evident during 2017 and 2018 (χ^2^ = 4.8, df = 1, *p* < 0.05) (Table 2).

The abundance of *Ae. aegypti* and *Ae. albopictus* was estimated by multiplying the number of eggs counted by the proportion of each species determined by laboratory rearing (Figure 2). Species abundance differed significantly among the sampling years (χ^2^ = 148.9, df = 2, *p* < 0.001) and was lower in the dry seasons compared to the rainy seasons in both villages (χ^2^ = 44.2, df = 1, *p* < 0.001). In Río Florido, the abundances of *Ae. aegypti* and *Ae. albopictus* were similar indoor, and *Ae. albopictus* was higher than *Ae. aegypti* in ovitraps placed outdoors (χ^2^ = 44.4, df = 1, *p* < 0.001). In Hidalgo, *Ae. aegypti* was significantly more abundant than *Ae. albopictus* in both indoor and outdoor ovitraps (χ^2^ = 44.4, df = 1, *p* < 0.001).

In Río Florido, the estimated numbers of *Ae. aegypti* increased by 3-fold between the dry and rainy seasons for ovitraps placed indoors (2356 and 7044 individuals, respectively) and increased by 3.2-fold for ovitraps placed outdoors (1764 and 5620 individuals in the dry and rainy seasons, respectively). Similarly, the seasonal increase in the estimated abundance of *Ae. albopictus* was 2.6-fold for ovitraps located indoors (2535 and 6653 individuals in the dry and rainy seasons, respectively) and 3-fold in ovitraps placed outdoors (2507 and 7657 individuals, respectively).

In Hidalgo, estimated numbers of *Ae. aegypti* increased by 3.4-fold between the dry and rainy seasons in indoor ovitraps (3740 and 12,755 individuals, respectively), and by 4.1-fold in outdoor ovitraps (3609 and 14,980 individuals, respectively). Similarly, the increase in the estimated numbers of *Ae. albopictus* was 2.9-fold in indoor ovitraps and 3.7-fold in outdoor ovitraps (Figure 2).

Interestingly, laboratory rearing of samples collected from ovitraps revealed that in Río Florido, the prevalence of *Ae. aegypti* was higher during the dry season and decreased in the rainy season, whereas *Ae. albopictus* was relatively scarce during the dry seasons but increased in prevalence in the rainy seasons (Figure 3a). Similarly, in terms of mean estimated individuals, *Ae. aegypti* predominated in the dry season and decreased in the rainy season when *Ae. albopictus* became dominant in Río Florido (Figure 4a).

In contrast, in Hidalgo, the prevalence of *Ae. aegypti* was higher during the dry season and most of the rainy season and decreased towards the end of the rains, whereas the prevalence of *Ae. albopictus* was low in the dry season and increased at the end of the rainy season (Figure 3b). *Ae aegypti* also predominated during the dry season and *Ae. albopictus* was only predominant at the end of the rainy season with lower mean values of estimated individuals than observed in Río Florido (Figure 4a,b).

### 3.2. Sampling in Surrounding Transects

A total of 84,047 *Aedes* spp. eggs was collected in ovitraps located along the transects surrounding Río Florido during the entire sampling period (Appendix A). The numbers of eggs collected varied significantly in 2018 (39,965 eggs), 2017 (25,594 eggs), and 2016 (18,488 eggs) (χ^2^ = 77.6, df = 2, *p* < 0.001). Overall, 83.7% of eggs (70,339 eggs) were collected during the rainy season samples and the remainder (16.3%, 13,708 eggs) in the dry season samples taken in surrounding transects (χ^2^ = 185.0, df = 1, *p* < 0.001). The number of eggs collected in the dry season was similar in 2016 and 2016–2017 (2827 and 2377 eggs, respectively), but increased markedly in the 2017–2018 dry season (8504 eggs). In contrast, a total of 15,661 eggs was collected in the 2016 rainy season, 23,217 eggs in the 2017 rainy season and 31,461 eggs during the 2018 rainy season.

In terms of the spatial distribution of samples, the number of eggs collected decreased significantly with increasing distance from the edge of the village, with 33,107 eggs collected in ovitraps at the edge of Río Florido, 30,552 eggs in ovitraps located at 50 m, and 20,388 eggs in ovitraps located at 100 m from the edge of the village (χ^2^ = 15.9, df = 2, *p* < 0.001) (Appendix A).

In the ovitraps placed along the transects surrounding Hidalgo, 67,542 eggs were collected (Appendix A), 20.0% less than the number of eggs collected from the transects surrounding Río Florido. The number of eggs collected each year varied from 35,005 eggs in 2018, 19,300 eggs in 2017, and 13,237 eggs in 2016 (χ^2^ = 77.6, df = 2, *p* < 0.001). Overall, 82.9% of eggs (55,981 eggs) were collected during the rainy seasons and the remainder (17.1%, 11,561 eggs) in the dry seasons (χ^2^ = 185.0, df = 1, *p* < 0.001). The number of eggs collected in the dry season varied from 6459 eggs in 2017–2018, compared to lower numbers in the dry seasons of 2016 (1834 eggs) or 2016–2017 (3268 eggs). In contrast, the number of eggs collected in each rainy season increased from 11,403 eggs collected during the 2016 rainy season, to 16,032 eggs and 28,546 eggs collected in the 2017 and 2018 rainy seasons, respectively.

The spatial distribution of eggs varied significantly in the transects with a higher number of eggs (31,014 eggs) collected at the edge of Hidalgo village, followed by the 100 m sample point (19,943 eggs) and the 50 m sample point (16,585 eggs) (χ^2^ = 15.9, df = 2, *p* < 0.001) (Appendix A).

*Aedes albopictus* was by far the dominant species in the zones surrounding both villages and also in the road transect between both communities. The abundance of *Ae. albopictus* was lower in the dry seasons compared to the rainy seasons in the zone around both villages (Figure 5).

In Río Florido, the estimated mean seasonal abundance of *Ae. albopictus* was 2318 ± 1205 individuals during the dry season (based on egg counts and laboratory rearing), which increased by 5.7-fold, with an estimated abundance of 13,192 ± 3705 individuals in the rainy season. In Hidalgo, the estimated mean seasonal abundance of *Ae. albopictus* during the dry season was 1817 ± 669 individuals with an increase of 4.4-fold in the rainy season (8027 ± 2439 individuals).

The estimated seasonal abundance of *Ae. aegypti* was very low in the area surrounding both villages (Figure 5). Surrounding Río Florido, the abundance of *Ae. aegypti* in the dry season was of 35 ± 31 individuals which increased 3.4-fold (120 ± 33 individuals) in the rainy season.

The estimated abundance of *Ae. aegypti* at the edge of Río Florido was 18 ± 16 and 66 ± 23 individuals in the dry and rainy seasons, respectively. These seasonal values were higher compared to seasonal averages of 12 ± 12 and 51 ± 34 individuals in ovitraps placed at 50 m, or 4 ± 3 and 8 ± 5 individuals at 100 m from the edge of the village in the dry and rainy seasons, respectively.

The abundance of *Ae. aegypti* was generally higher in the zone surrounding Hidalgo compared to the abundance recorded in Río Florido. During the dry season, the estimated mean abundance of *Ae. aegypti* was 106 ± 8, which increased 4.4-fold to 470 ± 104 individuals in the rainy season. The estimated number of individuals decreased with distance from the edge of Hidalgo village (90 ± 31 and 353 ± 116), at 50 m (15 ± 15 and 36 ± 15) and 100 m (11 ± 11 and 73 ± 30) from the edge of the village in the dry and rainy season, respectively (Figure 5).

A total of 14,073 *Aedes* spp. eggs was collected in the road transect between the two villages, of which 11,973 eggs (85.1%) were obtained during the rainy season and the remainder (2100 eggs, 14.9%) in the dry season. *Aedes albopictus* was the predominant species representing 100% of laboratory-reared individuals during the dry season and 99.7% in the rainy season, whereas *Ae. aegypti* represented just 0.3% of reared individuals during the rainy season. The presence of *Ae. aegypti* was likely related to the proximity of inhabited houses at a distance of ~100 m from some sampling points close to the road.

Clear correlations were detected between intradomiciliary and peridomiciliary seasonal egg counts for both villages (t = 8.05, df = 10, *p* < 0.001), with adjusted coefficient of determination (R^2^_adj_) values of 0.92 and 0.94 for Río Florido and Hidalgo, respectively (Figure 6). Similarly, strong correlations were detected between intradomiciliary and peridomiciliary estimated abundance of adults for both *Ae. aegypti* and *Ae. albopictus* in both villages and both seasons, with R^2^_adj_ values that varied between 0.68 and 0.99 depending on site and season (Appendix A).

## 4. Discussion

A detailed ovitrap-based evaluation of *Aedes* spp. populations was performed in two rural villages in southern Mexico over a three-year period. *Aedes albopictus* was the dominant species in the zones surrounding both villages (Figure 5) and along the road transect. There was a markedly lower coexistence of *Ae. aegypti* and *Ae. albopictus* in the surrounding zones, which increased at the edge of the villages near to houses, and was highest in the areas of housing. These results are consistent with studies in which *Ae. albopictus* was associated with areas of dense vegetation and coexists with *Ae. aegypti* in transitional zones that are typically found in peri-urban areas [46,47]. The predominance of *Ae. albopictus* in the zones surrounding our study villages is likely to reflect a combination of the preference of this species to oviposit in natural sites over artificial containers, and its ability to feed on a range of different animal species [6,48,49].

Differences in the population dynamics of both species were recorded between villages. These differences could be related to the niches and environmental differences present in each village and the contiguous surrounding zone, and differences in anthropogenic activities involving vector control measures. Oviposition was lower during the dry season and increased in both villages during the rainy season. However, in Hidalgo, 1.6-fold more eggs were collected than in Río Florido and more eggs were laid in outdoor ovitraps (58.1%) compared to those placed indoors (41.9%).

Community engagement activities related to the SIT research project had been performed since 2017 in both villages. These activities focused on the education of the villagers and their children on the biology and ecology of *Ae. aegypti*, its role in disease transmission, and habitat elimination activities targeted at potential oviposition sites in each village. Although the community engagement activities were similar in both villages, average egg counts in outdoor ovitraps in Río Florido were significantly lower than outdoor ovitraps in Hidalgo. This may have been due to differences in the behavior of local inhabitants [50]. The inhabitants of Río Florido had also previously participated in community engagement activities for dengue control in the framework of a nearby confined-field trial of genetically modified *Ae. aegypti* from 2008 to 2011 [51], so they may have been more aware of the need for safe disposal of potential oviposition containers (bottles, tires, plastic trays, etc.) than the inhabitants of Hidalgo.

Although the following data are beyond the study period interval, they illustrate clear differences in the behavior of the inhabitants of each village. Municipal public health workers removed solid waste on three occasions during 2019 and eliminated more than 3000 potential oviposition sites in each village. Although the amounts of removed solid waste were similar in both villages, an entomological survey performed by state vector control workers in early 2020 noted several differences in the risk of vector-borne disease in each of the villages. In Río Florido, 23.3% of the houses and just 4.1% of artificial containers (empty bottles, vehicle tires, etc.) had *Aedes* larvae and pupae present, but most water tanks had larvivorous fish present and most artificial containers (bottle, tires etc.) had been protected from rainfall and were not suitable for mosquito oviposition. In contrast, in Hidalgo, 65.5% of the houses and 8.9% of controllable containers had *Aedes* larvae and pupae present, most solid waste was casually discarded outdoors, and very few water tanks had larvivorous fish or larvicide treatments present (E.P. Contreras-Mejia, State Health District VII, personal communication). As a result, the higher numbers of eggs collected in outdoor ovitraps in Hidalgo (Table 2) may be related to the abundance of household trash found around houses that formed oviposition sites that could benefit both species [6,52], although they differ in microhabitat preferences [3,53]. In this respect, SIT-based population suppression will likely be improved by engaging the village inhabitants in a continuous campaign of habitat elimination and larvicidal treatment of water tanks. Such activities could have a two-fold benefit in both reducing the number of sterile males required for suppression of the *Ae. aegypti* population and reducing the availability of oviposition sites for *Ae. albopictus*, with the associated benefits for human health. Our previous outreach activities in these communities suggests that the majority of villagers are willing to engage in vector elimination campaigns over periods of several months [51].

*Aedes aegypti* was the dominant species during the dry season and the start of the rainy season in both villages, whereas populations of *Ae. albopictus* were low in the dry season, but subsequently increased and became dominant during the rainy season (Figure 4a,b). Environmental conditions, biological, and anthropogenic factors are involved in population fluctuations of *Ae. aegypti*. Low humidity (dry season) and high temperature conditions are generally more favorable for *Ae. aegypti* because its eggs are resistant to desiccation [54]. This species is also capable of using a diversity of artificial oviposition sites, is often present inside buildings and is mostly anthropophilic [53,55]. In contrast, *Ae. albopictus* eggs can enter diapause during periods of low temperature [56,57], but the eggs are less tolerant of desiccation compared to those of *Ae. aegypti* [54]. However, the marked ecological plasticity of *Ae. abopictus* allows it to take advantage of a wide range of natural breeding sites [6]. Rainy season conditions result in an increase in the number of natural oviposition sites available during this period in habitats with abundant vegetation, which are often exploited by *Ae. albopictus* and less frequently by *Ae. aegypti* [6,46,47,48,49,53].

The greater dominance of *Ae. aegypti* and smaller increase in the *Ae. albopictus* population observed in Hidalgo compared to that of Río Florido may be related to factors such as the management of solid waste discarded outdoors and the presence of few treated water drums and water tanks rather than differences in the natural vegetation, as Hidalgo had 62% of natural vegetation cover surrounding the village compared to 49% in the area surrounding Río Florido (Figure 1)**,** where the *Ae. albopictus* population was clearly dominant during the rainy season. We suggest that the persistence of the *Ae. aegypti* population in Hidalgo during both the dry and rainy seasons may be related to the abundance of household trash abandoned in this village, and the presence of water tanks without larvivorous fish or larvicidal granules, which likely favored oviposition by this species. *Aedes aegypti* also tends to engage in skip oviposition to a greater degree than *Ae. albopictus* [58], which as a risk-reduction strategy may have favored the persistence of *Ae. aegypti* in the village with a greater number of marginal trash-based oviposition sites in the rainy season. In contrast, household trash containing small volumes of water is often avoided for oviposition by *Ae. albopictus* [52], especially if already occupied by mosquito larvae of another species [59].

Although the preferred breeding sites of *Ae. albopictus* are usually outdoor sites [60,61,62,63,64,65], this species has also been recorded using indoor containers for oviposition, particularly in some Asian countries [66,67]. Indeed, we collected *Ae. albopictus* eggs in indoor ovitraps throughout the sampling period in both villages, and at high prevalence during the rainy season, underlining the ability of this species to adapt to local ecological conditions, which could indicate plasticity in the reproductive and host-seeking behaviors of the *Ae. albopictus* population that allowed it to exploit oviposition and even potential feeding opportunities inside houses [6]. As the intradomiciliary and peridomiciliary populations of both *Ae. aegypti* and *Ae. albopictus* were strongly correlated in both villages and both seasons (Figure 6, Appendix A), future studies in these villages could omit or greatly reduce intradomiciliary sampling without a significant loss of information. Access to people’s dwellings on a regular basis for mosquito sampling can be problematic, as house occupants may be out when vector control workers call, or may be reluctant to grant access if alone in the house, an issue that is exacerbated in regions currently experiencing increasing levels of violence, such as Mexico.

The implementation of a SIT-based program with an area-wide approach for *Ae. albopictus* that is well established in rural and peri-urban zones, can be more complicated and require more effort than programs targeted at isolated *Ae. aegypti* populations within clearly defined zones of human habitation. Compared to *Ae. aegypti*, releases of *Ae. albopictus* sterile males would have to be performed over a larger area, which should include human housing and industrial areas and surrounding zones of natural habitat. The high availability of natural breeding sites [6] and the immigration of congeners from surrounding areas would likely hinder efficient suppression of *Ae. albopictus* populations.

Under ideal conditions, the most conducive approach may be to undertake an area-wide SIT-based program targeted at both species. However, this would markedly increase the cost of the program. Given this, the pragmatic approach would be to initially target *Ae. aegypti*, as it is the most important vector of dengue in this region. If *Ae. albopictus* occupies the niche left by *Ae. aegypti* [68] and becomes the main disease vector in SIT-treated areas, additional efforts in habitat elimination and larvicidal treatment of water sources would have to be made for the suppression of this species. The fact that population decline of *Ae. aegypti* can occur rapidly in the presence of relatively low densities of *Ae. albopictus* indicates that multiple factors are involved in this phenomenon [69]. Such factors include local environmental conditions and changes in aquatic habitats [70], the presence of species-specific parasites [71,72], and asymmetrical reproductive interference resulting in sterility of *Ae. aegypti* females that mate with *Ae. albopictus* males [73], although the prevalence of heterospecific mating decreases over time [74]. *Aedes albopictus* larvae also outcompete larvae of *Ae. aegypti* in habitats with a sparse food supply [75]. The possibility of niche replacement highlights the importance of multi-year baseline studies such as reported here, which will allow us to test whether future SIT-based suppression of *Ae. aegypti* results in an increase in the population size and distribution of *Ae. albopictus*.

## 5. Conclusions

A three-year program of sampling ovitraps revealed that *Ae. albopictus* was the dominant species surrounding two rural villages in southern Mexico, whereas *Ae. aegypti* was dominant within the villages for most of the year. The abundance of both species increased significantly during the rainy season in both villages, but the prevalence of the *Ae. albopictus* population increased markedly and became dominant during the rainy season. The greater dominance of *Ae. aegypti* in Hidalgo village may be due to the presence of abandoned household trash and water tanks without appropriate treatments. Spatially isolated populations of *Ae. aegypti* in village communities are likely to favor successful application of an area-wide SIT-based suppression program serving as an ecological island, but it will be necessary to monitor *Ae. albopictus* populations due to possible niche expansion by this species in the absence of *Ae. aegypti*, and the corresponding implications for dengue virus transmission among village inhabitants.

## Figures and Tables

**Figure 1 insects-12-00058-f001:**
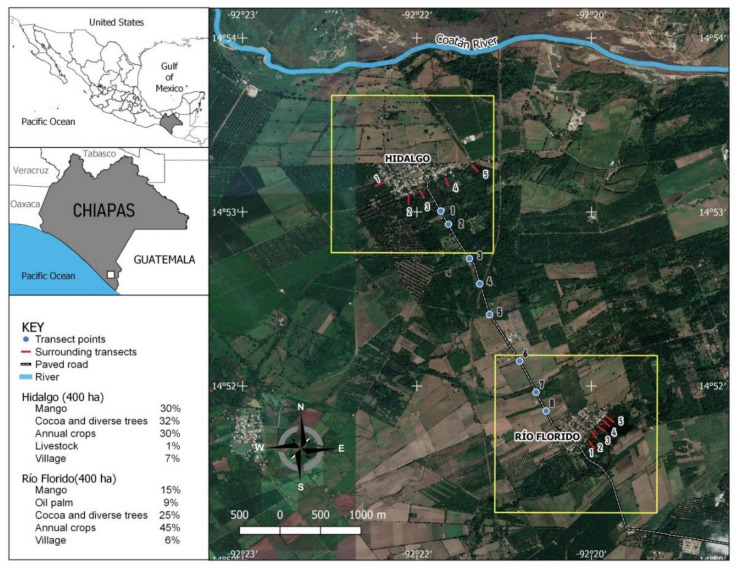
Location of the Hidalgo and Río Florido study villages in Tapachula municipality, Chiapas State, southern Mexico.

**Figure 2 insects-12-00058-f002:**
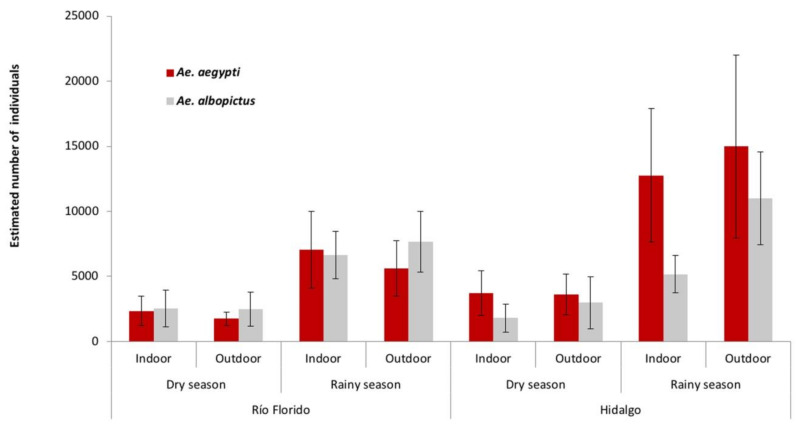
Mean (+SE) estimated abundance of *Aedes aegypti* and *Ae. albopictus* based on egg counts from indoor and outdoor ovitraps in the dry and rainy seasons during a three-year study in Río Florido and Hidalgo villages. Abundance was calculated as the product of number of eggs per location of ovitrap in village by season and the proportion of each species present in laboratory-reared samples.

**Figure 3 insects-12-00058-f003:**
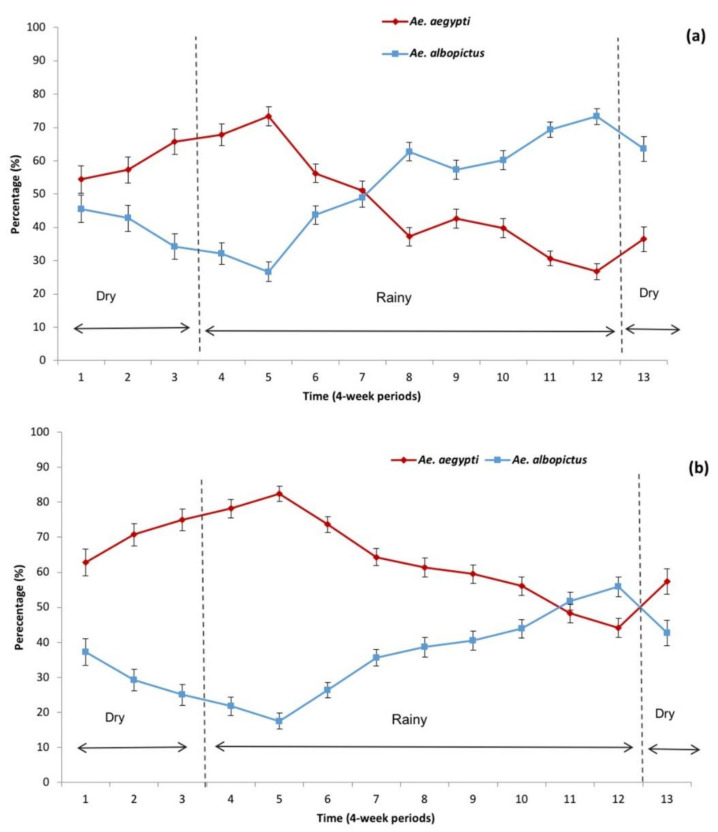
Mean percentage (+SE) of *Aedes aegypti* and *Ae. albopictus* reared from eggs collected from indoor and outdoor ovitraps per 4-week period during a three-year study in: (**a**) Río Florido, (**b**) Hidalgo village. Four-week periods were based on the arrival of the rainy season (>100 mm precipitation/4-week) in each year of the study.

**Figure 4 insects-12-00058-f004:**
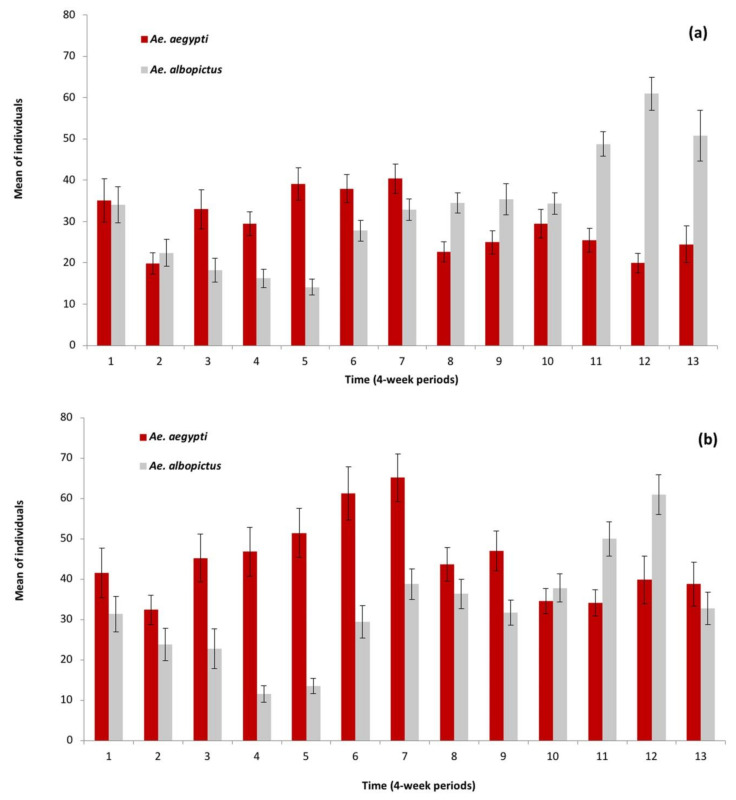
Mean (+SE) estimated abundance of *Aedes aegypti* and *Ae. albopictus* considering both in and outdoor traps per 4-week period during three years in: (**a**) Río Florido, (**b**) Hidalgo. Abundance is the product of the number of eggs collected from ovitraps, egg viability (hatching), and the proportion of each species identified in laboratory-reared samples.

**Figure 5 insects-12-00058-f005:**
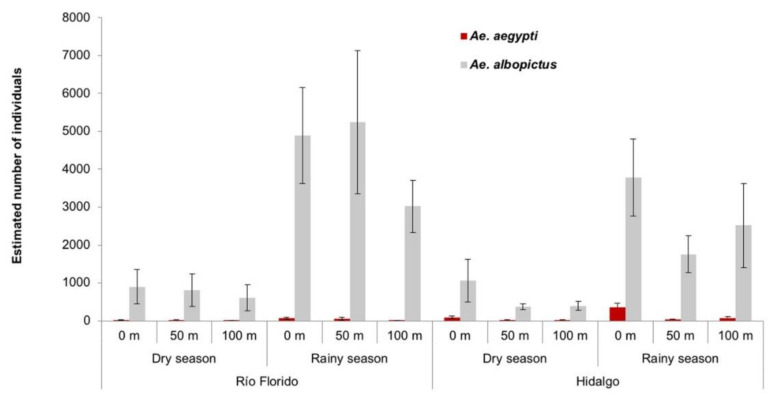
Mean (+SE) estimated abundance of *Aedes aegypti* and *Ae. albopictus* in the zones surrounding Río Florido and Hidalgo villages in the dry and rainy seasons during a three-year study. The estimated abundance is the product of number of eggs per location of ovitrap in transect by season and the proportion of each species present in laboratory-reared samples.

**Figure 6 insects-12-00058-f006:**
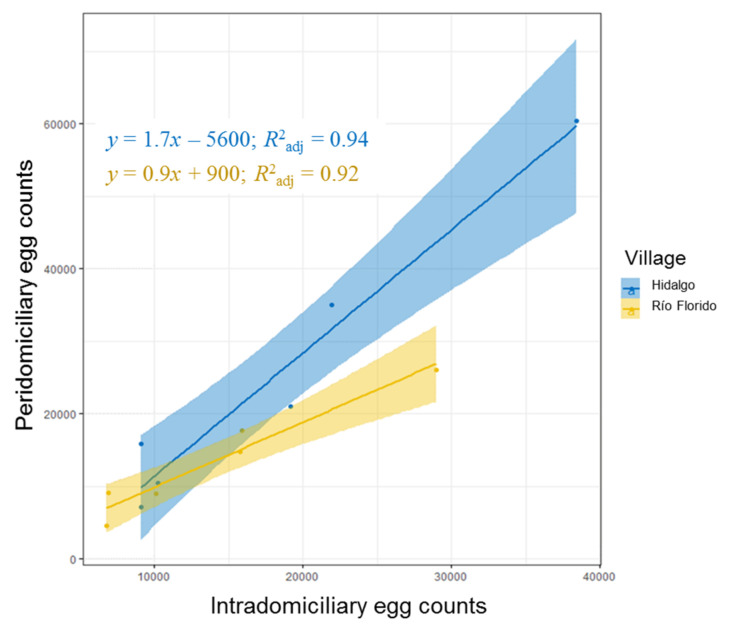
Correlations between seasonal eggs counts obtained from intradomiciliary and peridomiciliary ovitraps placed in Hidalgo and Río Florido over the three-year study. Shaded regions around fitted lines indicates the 95% confidence interval of each correlation.

**Table 1 insects-12-00058-t001:** Total number of *Aedes* spp. eggs collected from indoor and outdoor ovitraps in Río Florido, number of eggs used for laboratory hatching and rearing studies, and prevalence of *Aedes aegypti* and *Aedes albopictus* determined by laboratory rearing from January 2016 to November 2018.

					*Aedes aegypti*	*Aedes albopictus*
Year and Season of Sample	Mean Rainfall (mm)	Location in Village	Total Eggs Collected in Ovitraps	Total Eggs Tested in Laboratory	Number of Larvae Reared (%)	Number of Larvae Reared (%)
2016 Dry	2.1	Indoor	5595	1781	408 (49.5)	417 (50.5)
		Outdoor	6801	1864	411 (52.4)	373 (47.6)
2016 Rainy	363.5	Indoor	15,773	8437	1436 (42.0)	1983 (58.0)
		Outdoor	14,749	7883	1145 (39.6)	1750 (60.4)
2016–2017 Dry	13.0	Indoor	4363	2601	685 (55.5)	550 (44.5)
		Outdoor	4462	2378	551 (49.6)	559 (50.4)
2017 Rainy	342.6	Indoor	15,916	7614	2785 (48.3)	2983 (51.7)
		Outdoor	17,749	8735	2558 (37.9)	4200 (62.1)
2017–2018 Dry	5.0	Indoor	13,860	5482	1834 (46.3)	2125 (53.7)
		Outdoor	11,322	5481	1335 (35.0)	2474 (65.0)
2018 Rainy	315.2	Indoor	28,957	13,302	5800 (55.8)	4601 (44.2)
		Outdoor	26,076	12,228	4464 (46.0)	5234 (54.0)
		Totals	165,623	77,786	23,412	27,249

**Table 2 insects-12-00058-t002:** Total number of *Aedes* spp. eggs collected from indoor and outdoor ovitraps in Hidalgo village, number of eggs used for laboratory hatching and rearing studies, and prevalence of *Ae. aegypti* and *Ae. albopictus* determined by laboratory rearing from January 2016 to November 2018.

				*Ae. aegypti*	*Ae. albopictus*
Year and Season of Sample	Location in Village	Total Eggs Collected in Ovitraps	Total Eggs Tested in Laboratory	Number of Larvae Reared (%)	Number of Larvae Reared (%)
2016 Dry	Indoor	6882	2001	751 (78.0)	212 (22.0)
	Outdoor	4924	1411	582 (77.7)	167 (22.3)
2016 Rainy	Indoor	19,166	9161	2549 (69.5)	1116 (30.5)
	Outdoor	20,996	8388	1711 (52.8)	1529 (47.2)
2016–2017 Dry	Indoor	5107	3090	920 (67.2)	449 (32.8)
	Outdoor	7978	3314	853 (59.2)	587 (40.8)
2017 Rainy	Indoor	21,906	9640	4540 (62.1)	2775 (37.9)
	Outdoor	35,046	12,223	4339 (46.4)	5003 (53.6)
2017–2018 Dry	Indoor	16,386	6767	2939 (64.5)	1621 (35.5)
	Outdoor	20,474	7112	2342 (49.2)	2414 (50.8)
2018 Rainy	Indoor	38,440	13,449	7912 (76.7)	2406 (23.3)
	Outdoor	60,407	14,018	6548 (65.5)	3442 (34.5)
	Totals	257,712	90,574	35,986	21,721

## Data Availability

All the data presented in this study are available in the file Supplemental Data.xlxs.

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
