# Peer review of "Population Dynamics of Aedes aegypti and Aedes albopictus in Two Rural Villages in Southern Mexico: Baseline Data for an Evaluation of the Sterile Insect Technique"

_insects, 2021, doi:10.3390/insects12010058_

Round 1

Reviewer 1 Report

The authors of this manuscript analyze the distribution of two aedes species in a specific area in Mexico based on oviposition traps. This manuscript is of limited interest to anyone not interested specifically in the distribution of those two species in Mexico.

A few minor comment:

Line 42: Is this sentence correct? The authors cite a manuscript about the distribution of both species.

Line 47, 54 and other places: full name at the beginning of a sentence.

Line 95: mm?

Line 157: provide software source

Lines 318-324: This paragraph belongs at the beginning of the results section.

Figures 3-4: the x-axis should include names of months

Reviewer 2 Report

Population dynamics of Aedes aegypti and Aedes albopictus in two rural villages in southern Mexico: baseline data for an evaluation of the sterile insect technique

This longitudinal study monitored ovitrap data in two isolated villages with the perspective of using those sites to test the effectiveness of SIT on Ae. aegypti. These type of data over several years provide a good starting point to understand temporal and spatial variations in the Aedes spp. populations in southern Mexico. The manuscript is well written, and the results are meaningful. It is important to understand how Ae aegypti and Ae albopictus populations fluctuate through the dry and rainy seasons in a tropical setting. The results also provide insight into the spatial dispersal of these species. There are some minor comments to improve our understanding of the methodology/results.

Line 78. It might be useful to clarify that the SIT program will target just Ae. aegypti at this point.

Line 87. Was there any mosquito control going on in the study sites?

Line 115. What was the basis for the decision of using 15 houses for ovitrap placement (1 indoor, 1 outdoor)?

Line 123. What was the distance from the village border of the transects at the outer edge of the villages? What was the rationale for this surveillance?

Line 126. Were there any human dwellings at the places where these ovitraps were placed (that would suggest local production of Aedes spp. vs. dispersal)? Provide a brief description of the places where these ovitraps were located (e.g., under a tree, etc.). Please provide the rationale for this surveillance activity (what did you expect to happen or reveal).

Line 143. Please provide a reference for the hatching method of placing eggs at 38C.

Line 156. Please specify what dependent variables and hypotheses were tested using GLMM. Because these are sequential samples (weekly) there is going to be a significant effect of temporal autocorrelation. Did you take that into account in the models, such as by using a repeated measures autoregressive covariance approach or similar?

Line 161. There is much repetition of data in Results already shown in tables and figures.

Line 162. Temperature and relative humidity measured “at the moment of sampling” were taken at the same time of the day or how were those measures standardized?

Tables 3 and 4 may me moved to Supplemental because Fig. 5 already represents those values.

Line 373. Please explain: …that could benefit Ae. aegypti more than Ae. albopictus.

Line 385. Please clarify: …habitats with abundant vegetation, such as those that proliferate in the rainy season, are more favorable for Ae. albopictus and less favorable for Ae. aegypti.

What would be the correlation between eggs indoors and outdoors (per species)? If there is a highly significant correlation, indoor sampling would not be necessary looking forward. Similarly, what is the correlation between aegypti and albopictus (per village, indoor/outdoor)? This information can be useful to understand coexistence between these species.

Line 388. The explanation whereby aegypti is favored by the existence of outdoor trash containers in comparison with albopictus is not clearly understood because, as explained by the authors, albopictus uses outdoor containers.
